# Influence of Puerperal Health Literacy on Tobacco Use during Pregnancy among Spanish Women: A Transversal Study

**DOI:** 10.3390/ijerph17082910

**Published:** 2020-04-23

**Authors:** Rafael Vila-Candel, Esther Navarro-Illana, Desirée Mena-Tudela, Pilar Pérez-Ros, Enrique Castro-Sánchez, Francisco Javier Soriano-Vidal, Jose Antonio Quesada

**Affiliations:** 1Department of Obstetrics and Gynaecology, La Ribera University Hospital, FISABIO, Crta. Corbera km 1, 46600 Valencia, Spain; rafael.vila@uv.es; 2Department of Nursing, Faculty of Nursing and Podiatry, University of Valencia, Jaume Roig, s/n, 46010 Valencia, Spain; francisco.j.soriano@uv.es; 3Department of Nursing, Catholic University of Valencia “San Vicente Mártir”, Espartero 7, 46007 Valencia, Spain; pilar.perez@ucv.es; 4Department of Nursing, Jaume I University, Av de Vicent Sos Baynat, s/n, 12071 Castelló de la Plana, Spain; dmena@uji.es; 5NIHR Health Protection Research Unit (HPRU) in Healthcare Associated Infections (HCAI) and Antimicrobial Resistance (AMR) at Imperial College London, Du Cane Road, London W12 0NN, UK; e.castro-sanchez@imperial.ac.uk; 6Department of Obstetrics and Gynaecology, Lluis Alcanyis Hospital, FISABIO. Crta Xàtiva, s/n, 46800 Valencia, Spain; 7Department of Clinical Medicine, Miguel Hernández University, Av de la Universitat d’Elx s/n, 03202 Elche, Spain; jquesada@umh.es

**Keywords:** Health literacy, pregnancy, tobacco use, tobacco smoking

## Abstract

*Background*: Despite the fact that tobacco use during pregnancy produces adverse perinatal effects, some women continue to smoke. Health literacy (HL) is essential for health outcomes in adults. However, little is known about HL in pregnant women or postpartum women. The study aimed to analyse the relationship between the degree of HL of women during the early puerperium and tobacco use during pregnancy. Methods: A multicentre, descriptive, cross-sectional study was carried out with women in the early puerperium in a region of eastern Spain, between November 2017 and May 2018. Their HL level was obtained using the Newest Vital Sign (NVS) tool. Multivariate logistic models were adjusted to estimate the magnitude of association with tobacco use in pregnancy. Odds ratios (OR) were estimated with a 95% confidence interval. Results: 193 were included in the total. 29.5% (57) of pregnant women smoked tobacco during pregnancy, with a smoking cessation rate of 70.1% (40) while pregnant. 42.0% (81) of pregnant women had inadequate or limited HL. A low level of HL was strongly associated with tobacco use, adjusted by catchment area and age of first pregnancy (LRT *p* < 0.001; ROC curve = 0.71, 95% CI: 0.64–0.79). Conclusion: A low HL is associated with tobacco consumption during pregnancy. Whether low HL reflects the wide constellation of already-known socioeconomic, political and commercial determinants of tobacco use, or whether incorporating HL support interventions strengthens tobacco cessation activities in pregnancy, warrants further research. Still, it should be considered as essential to understanding the health disparities related to its consumption.

## 1. Introduction

Tobacco use has been identified as the most important cause of preventable death, adversely affecting the cardiovascular and respiratory systems and perinatal health [1]. The World Health Organisation advises pregnant women to abstain from tobacco use as there is no safe threshold [2]. Indeed, different authors have reported that tobacco use produces a wide range of adverse perinatal outcomes such as the increased risk of abortion, foetal loss, preterm delivery, low birth weight, premature rupture of membranes, premature placental abruption, or delayed foetal growth [3,4]. Despite extensive information being available on the risks of tobacco smoking for the foetus during pregnancy, some women continue to smoke [5]. Most countries lack current data on the prevalence of tobacco use during pregnancy [6]. Estimates suggest, however, that 1.7% of pregnant women worldwide are smokers. In Europe, it is estimated that 8.1% of the pregnant population smokes, a figure considerably higher than the global average [6]. In Spain, the figures are even more discouraging, ranging between 30–45% of women smoking tobacco in early pregnancy [7,8], although around 40% of those usually quit within the first trimester of pregnancy [7].

To analyse tobacco use during pregnancy, different methods have been used to measure tobacco exposure, including self-administered questionnaires, measurements of expired carbon monoxide or cotinine concentration in urine [9,10]. 

Health literacy (HL) relates to the user’s knowledge and skills in decision-making in a medical and social context [11]. These skills include reading, writing, making calculations, communicating, searching for information, using electronic technologies and solving problems, all of which are essentially personal and social skills for navigating the health system [12,13]. 

Currently, there are different standardised and validated instruments available for assessing HL, mostly in English [14] and focused on North American citizens. In Europe, there is a collaboration to standardise a valid questionnaire [11] applicable to European people [15]. In Spanish, there are previously validated and widely recognised questionnaires, such as Short Assessment of Health Literacy for Spanish Adults (SAHLSA_50) [16,17,18], Newest Vital Sign (NVS) [19,20,21,22] and Single Item Literacy Screener (SILS) [23]. 

There is currently controversy regarding the routine use of population screening for HL. Some experts recommend considering the entire population as having a low HL level [24], claiming that routine screening of HL has not shown benefits and could have undesired effects. On the contrary, different professional organisations [11,25] promote HL screening to reach the largest possible population and provide understandable and accessible information, regardless of the level of HL. 

Multiple bio-psycho-social and economic factors influence both decisions to start and quit smoking tobacco [16,26,27,28]. HL is emerging as a fundamental mediator for such decisions [29]. Limited or poor HL appears to be a crucial factor in smoking from adolescence [30] to adults over 50 years [31].

Different studies on possible demographic predictors related to smoking have been published internationally, although the benefit of interventions carried out during pregnancy is still inadequate [5,32,33,34]. Little is known about the impact of health literacy on women’s health and tobacco use in pregnancy.

Therefore, this study aimed to analyse the relationship between the HL of puerperal women and tobacco consumption during pregnancy.

## 2. Methods

### 2.1. Design, Population, and Sample

A multicentre, descriptive, cross-sectional study was carried out through a survey in women in the early puerperium (48 h postpartum), who gave birth at the Lluis Alcanyís Hospital (LLAH) and La Ribera University Hospital (LRUH), both abutting at the southern area of Valencia (Spain). These hospitals serve a population of 250,000 and 210,000 inhabitants, respectively, with an annual average of 1800 (LRUH) and 1100 (LLAH) births in the year the study was carried out. 

Postpartum women during November 2017 and May 2018 were included. Those women with visual and hearing disabilities that prevented them from completing the questionnaire, women with neuro-cognitive pathology, dementia or diagnosed mental health disorders, those refusing to participate in the study, those with a language barrier that hindered understanding, or those under 18, were excluded. 

Assuming a prevalence of 20% tobacco use in the adequate HL group, admitting a literacy ratio of 2:1 (adequate: inadequate) with 80% statistical power and a type I error of 0.05%, to detect a prevalence of 40% in the inadequate HL group, 189 participants were necessary for the study. To account for 5% of possible attrition, we estimated a sample of 198 participating women. 

It should be highlighted that the only behaviour considered by interviewers and participants was tobacco smoking, no data was gathered regarding marijuana or other tobacco-related products (e-cigarettes, snus, cigars, chewing tobacco, water pipes, etc.)

The study was carried out according to the principles of the Declaration of Helsinki for all medical research. The study was approved by the LRUH Ethics (HULR20170917) and Research Committee on 27/09/17.

### 2.2. Data Collection 

Prior to data collection, written consent was requested from each participant. Systematic sampling was carried out for the recruitment process, conducting interviews every 7 days with women admitted to the maternity ward on each of the days at each hospital. Data were obtained following two different methods: interviews (sociodemographic, obstetrical and level of HL variables) and electronic health records review (variables related to tobacco use).

During the immediate puerperium (24–48 h postpartum), an interview was carried out with each woman to analyse their level of HL and collect different sociodemographic variables (age, country of origin, marital status, occupation and activity of the pregnant woman and her partner, and education level) and obstetric characteristics (gestational age, parity, type of delivery and type of obstetric risk).

The participants’ HL was estimated using the *Newest Vital Sign* (NVS) tool. The tool evaluates reading comprehension and numerical skills based on six questions about the label of the nutritional composition of an ice-cream. The score is continuous quantitative on 6 possible points, corresponding to inadequate (0–1 points), limited (2–3 points) and adequate (4–6 points) health literacy. This tool has been validated for the Hispanic population residing in the United States. It has high sensitivity, but it may misclassify people with adequate literacy [35]. This tool has also been validated for the Spanish speaking population, with moderate reliability (Cronbach α = 0.69) [22].

During prenatal follow-up, midwives in Primary Care interviewed women regarding tobacco use in each of the trimesters, and it was registered in their electronic health records: (1) tobacco consumed during pregnancy; (2) the number of cigarettes per day (mean self-reported cigarettes smoked per day); (3) in case of quitting, in which trimester it took place.

### 2.3. Statistical Analysis

A descriptive analysis of all variables was performed by calculating frequencies of qualitative variables, and the minimum, maximum, mean and standard deviation values for quantitative variables. 

The factors associated with HL measured by the NVS screening tool were analysed, as well as tobacco use during pregnancy using contingency tables, applying the Chi-Square test or Fisher’s exact test for qualitative variables, and comparison of mean values for the quantitative ones, applying the Student’s *t*-test.

Multivariate logistic models were adjusted to estimate the magnitude of association with tobacco consumption in pregnancy. Odds Ratios (OR) were estimated along with their 95% CI. A stepwise variable selection procedure was performed based on the AIC (Akaike Information Criterium). Goodness-of-fit indicators and predictive indicators, such as the ROC curve, are shown. Analyses were performed using SPSS v.25 (IBM Corp. Released 2017. IBM SPSS Statistics for Windows, Version 25.0. Armonk, NY, USA) and R v.3.6.0 software (R Core Team (2019). R Foundation for Statistical Computing, Vienna, Austria), for a statistical significance <0.05.

## 3. Results

Out of 200 women who were informed about the study, 7 women (3.5%) did not wish to or could not participate for different reasons: 4 of them (57.2%) did not wish to participate, and 3 (42.8%) presented a language barrier. The final sample included 193 women (96.5%).

The mean age of the women surveyed was 32.9 (SD 5.4), with the mean age of the first pregnancy being 29.5 (SD 5.6) years. 53.3% (103) of the women of the sample were primiparous. The mean gestational week at delivery was 39.2 (SD 1.4). 29.5% (57) of the women smoked while pregnant, with a mean of 7.2 (SD 4.2) cigarettes per day at the beginning of the pregnancy. Table 1 shows the rest of socio-demographic, clinical and obstetric characteristics of the sample with the grouped variables. We observed that the prevalence of smokers was significantly higher in the catchment area of La Ribera (*p* = 0.003), in single-divorced-separated (*p* = 0.003) and in the low education level group (*p* = 0.017), as well as in limited or inadequate HL group (*p* = 0.004).

The smoking cessation rate during pregnancy was 70.2% (40). Pregnant women who did not quit during pregnancy had a mean tobacco use of 8.35 (SD 4.8) cigarettes per day, compared to those who quit smoking, with 6.7 (SD 3.9) cigarettes per day. The differences found were not statistically significant between the mean tobacco consumption and smoking cessation rate (*n* = 57; *p* = 0.267). Smoking cessation rate during the first trimester was 36.8% (21), in the second, 22.8% (13), and in the third, 10.5% (6). 29.8% (17) of the pregnant women did not quit. 

Table 2 shows the tabular forms of quantitative variables by level of HL. As for HL, the mean score for the NVS scale was 3.7 (SD 1.6) points. These values were categorised into inadequate HL (9.3% (18)), limited HL (32.6% (63)) and adequate HL (58.0% (112)). Due to scarce number of cases in the Inadequate HL category, Inadequate and Limited HL were combined, in order to calculate chi-square. 

Table 3 shows the percentage of level of HL category for quitters/no quitters. Of the 24 women with Adequate HL, 16 (67%) quit. Regarding those with inadequate or limited HL, a larger percentage quit (24% of 33% or 72%). 

The degree of health literacy and the rest of variables were presented in Table 4. The results suggest that the higher the level of education, the higher the HL (*p* < 0.001). Salaried workers had a higher level of HL than those unemployed (*p* = 0.041), just like women working in administration or in the health sector, who had a higher HL than those unemployed or those working in the service sector (*p* = 0.008). Non-smokers had a higher HL than smokers (*p* = 0.004). Women with low-risk pregnancies had a higher HL than those at high risk (*p* = 0.016). 

Finally, a multivariate logistic model was constructed with the variables that showed statistical significance and clinical relevance to determine the magnitude of association with smoking during pregnancy with the different explanatory variables (Table 5). Thus, we observed that limited and or inadequate literacy was strongly associated with tobacco use, adjusted by catchment area and age of first pregnancy, obtaining a model that fits well the data well (LRT *p* < 0.001; ROC curve = 0.71, 95% CI: 0.64–0.79).

## 4. Discussion

This study presents the evaluation of HL in women during the puerperium in Spain, and according to their tobacco consumption. The results suggest that the level of HL is inversely associated with tobacco consumption, as has been observed in other populations [36], but not so much as in pregnant or postpartum women.

Moreover, a low HL has been related to being unmarried, having a low education level, being unemployed, being a smoker, not quitting tobacco during gestation and having a high-risk pregnancy. The relationship between socioeconomic status, education level and other factors, both in pregnant women and in adults, have already reported tobacco use, and they coincide with our results [5,37,38,39]. Moreover, HL does not necessarily reflect what would otherwise be considered general literacy, given the numbers of women with limited educational attainment showing adequate HL and the finding that some women with high educational attainment showing limited or inadequate HL [11,13].

All these factors could be related to the intergenerational transmission of health inequalities, as other studies have already found [40].

The proportion of pregnant women who quit smoking during the first trimester and before delivery coincides with previous studies [7], although no statistically significant differences were found between the level of HL and smoking cessation during pregnancy. The associations are still significant after adjusting for sociodemographic and clinical characteristics known to be associated with the prevalence of tobacco use. Therefore, HL seems to be an associated factor with tobacco use, and pregnant women with a lower HL may be more likely to continue to smoke during pregnancy. 

Smoking during pregnancy has been associated with sociodemographic factors such as lower economic level, high parity, having no partner or having a partner who smokes, having a lower educational level and having a higher consumption of cigarettes the day before pregnancy [5]. Additionally, there are motivational factors involved such as not believing that tobacco affects the health of the foetus or newborn or having smoked in previous pregnancies [41]. Moreover, determining whether or not the women’s motivational aspects were the main reason for quitting smoking, or as a result of the process, is difficult and was not considered in our study. 

Few studies have investigated the relationship between HL and smoking; however, a study conducted in Louisiana (USA) in 2001 estimated the relationship between HL and smoking risk knowledge and attitudes among low-income pregnant smokers [39]. The results of these authors indicated that a low HL was associated with lower knowledge of the risk of smoking and less negative attitudes related to smoking. The profile of the woman included in the study, from 12 to 43 years of age with low economic resources, is far from the characteristics of the women in our study. In a systematic review in 2016, the relationship between these variables was evident in adolescents and young adults [36]. 

The results observed in our study provide evidence that low HL can be an important and independent association factor with tobacco use during pregnancy, contrary to what happens in other groups [42,43]. The results also show that HL is related to sociodemographic characteristics and that it can be an important element to explain the health disparities in pregnant women [14,43,44,45], since we have been able to observe that the same determinants that mark tobacco use influence HL, and that it is possible that HL may be a link between these groups. 

Low HL could also hinder self-efficacy in complying with the therapeutic follow-up of the different pharmacological approaches or behavioural techniques to quit smoking [46,47], and that stressful environments that favour tobacco consumption may include factors that drive low HL. Perhaps addressing or mitigating some of these common determinants could offer a benefit for HL and tobacco use, or even prenatal planning that would facilitate smoking to be stopped before gestation in a planned manner. This hypothesis should be trialled in future studies since it is beyond our main objective. 

HL plays an important role in influencing how smokers respond to different risk messages [48]. It is crucial to tailor health promotion messages to women with low HL, as has already been proposed by other authors [46,49]. Therefore, our findings highlight the importance of increasing awareness of the impact of HL on suboptimal health behaviours, including tobacco use during pregnancy and improving the training of healthcare workers to communicate clearly about the risks of smoking [43]. Once we have observed the association between HL and tobacco consumption, we could suggest that by providing information according to the level of HL, we could make it more feasible for pregnant women to quit smoking. Healthcare providers should be trained to communicate clearly with patients about the health consequences of smoking, for example, use plain language, visual aids (e.g., pictographs), and techniques such as the teach-back method to convey smoking health risks [42]. Different strategies have been reported in the literature, such as the use of simple language, individual teaching, different teaching methods, and electronic tools, all with positive results. Any specific communication or training intervention in low HL groups would improve the individual’s understanding and self-care, as different randomised clinical trials have concluded [50,51,52,53].

### 4.1. Limitations

Firstly, our results are based on the validity of the responses to the self-reported questionnaire. We are aware that measuring tobacco use through a self-reported questionnaire is a limitation and that it would have been better also to carry out urine cotinine or carbon monoxide testing. Consequently, the smoking rate observed may indeed underestimate the true ratio, due to the potential for socially desirable responses by the participants. The factors most closely related to concealing one’s smoking status have to do with the timing, and the quantity of tobacco consumed [7]. Some authors have observed the deeply associated stigma with smoking in public while pregnant; therefore, our figures could be even more concerning [31]. Secondly, the study design only allows us to report on associations and not causal relationships. Therefore, future studies should investigate possible underlying bio-psycho-social mechanisms between low HL and tobacco consumption during gestation and its cessation. In the third place, the measurement of HL was carried out during the early puerperium (48 h postpartum). We have assumed that HL is demonstrated consistently from the beginning to the end of pregnancy and that during pregnancy, no substantial modification of HL is expected. This methodology has already been used by different authors [37,46]. To our knowledge, no studies exist evaluating initial and final HL levels without intervention in pregnant women, so future studies should look into this. Finally, the economic status was not available as a variable, but we measured the education level, occupation and activity, which are also socioeconomic status indicators in line with other authors [31].

### 4.2. Strengths

Probabilistic sampling was used to select the study population. Moreover, the estimated sample size was reached, meeting the sample representativeness criterion, and the NVS tool is available and freely accessible, both in Spanish and English. For all these reasons, the study demonstrates its translational interest, since it prospectively identifies the population to which we can adapt the information to achieve smoking cessation.

## 5. Conclusions

A low HL is associated with tobacco consumption during pregnancy. Whether low HL reflects the wide constellation of already known socioeconomic, political and commercial determinants of tobacco use, or whether incorporating HL support interventions strengthens tobacco cessation activities in pregnancy, warrants further research. Still, it should be considered as essential to understanding the health disparities related to its consumption.

## Figures and Tables

**Table 1 ijerph-17-02910-t001:** Clinical and sociodemographic characteristics of the sample by smoker status (*N* = 193).

Variable	Total		SMOKER				
			YES		NO		
			*n* = 57		*n* = 136		
	*N*	%	*N*	%	*N*	%	*p*-Value
CATCHMENT AREA							
Ribera	145	75.1	51	89.5	94	69.1	0.003 *
Xativa-Ontinyent	48	24.9	6	10.5	42	30.9	
MARITAL STATUS							
Single-divorced-separated	74	38.3	31	54.4	43	31.6	0.003 *
Married	119	61.7	26	45.6	93	68.4	
EDUCATION LEVEL							
No studies or Primary Ed.	58	30.1	24	42.1	34	25.0	0.017 *
VET	31	16.1	11	19.3	20	14.7	
HNC	49	25.4	15	26.3	34	25.0	
Bachelor’s degree	29	15.0	3	5.3	26	19.1	
University degree	26	13.5	4	7.0	22	16.2	
OCCUPATION							
Salaried	114	59.1	31	54.4	83	61.0	0.689
Unemployed	48	24.9	16	28.1	32	23.5	
Others (Student. SE)	31	16.1	10	17.5	21	15.4	
SECTOR							
Administration	43	23.0	14	24.6	29	21.3	0.361
Unemployed	62	33.2	23	40.4	39	28.7	
Healthcare	22	11.8	4	7.0	18	13.2	
Services	21	11.2	4	7.0	17	12.5	
Others (Industry. Education)	39	20.9	11	19.3	28	20.6	
COUNTRY OF ORIGIN							
Spain	177	91.7	52	91.2	125	91.9	0.875
Other	16	8.3	5	8.8	11	8.1	
PARTNER’S OCCUPATION							
Entrepreneur	21	10.9	6	10.5	15	11.0	0.943
Salaried worker	157	81.3	46	80.7	111	81.6	
Others	15	7.8	5	8.8	10	7.4	
PREGNANCY RISK							
Low risk	154	79.8	42	73.7	112	82.4	0.171
High risk	39	20.2	15	26.3	24	17.6	
HEALTH LITERACY							
Adequate	112	58.0	24	42.1	88	64.7	0.004 *
Limited or inadequate	81	42.0	33	57.9	48	35.3	

* *p* < 0.05 Chi-Square or Fisher’s exact tests. VET: Vocational Education and Training; HNC: Certificate of Higher Education (HNC); SE: Self-employed; NVS: Newest Vital Sign.

**Table 2 ijerph-17-02910-t002:** Quantitative variables as per level of HL, ANOVA test.

Variable	*N*	Mean	SD	*p*-Value
NUMBER CIGARETTES/DAY				
Adequate HL	24	7.0	4.4	0.821
Limited or inadequate HL	33	7.3	4.1	
AGE				
Adequate HL	112	33.3	4.9	0.255
Limited or inadequate HL	81	32.3	6.0	
PARITY				
Adequate HL	112	1.6	1.0	0.572
Limited or inadequate HL	81	1.7	0.9	
AGE FIRST PREGNANCY				
Adequate HL	112	30.0	4.9	0.177
Limited or inadequate HL	81	28.9	6.4	
GESTATIONAL AGE				
Adequate HL	112	39.2	1.4	0.800
Limited or inadequate HL	81	39.1	1.4	

HL: Health literacy; SD: Standard deviation.

**Table 3 ijerph-17-02910-t003:** Percentage of level of HL for quitters/no quitters (*n* = 57).

Variable	SMOKING CESSATION DURING GESTATION				
	Quits		Does Not Quit		
	*n* = 40		*n* = 17		
	*N*	%	*N*	%	*p*-Value
HEALTH LITERACY					
Adequate HL	16	66.7	8	33.3	0.771
Inadequate or limited HL	24	72.7	9	27.3	

HL: Health Literacy by Newest Vital Sign.

**Table 4 ijerph-17-02910-t004:** Relationship between the degree of HL and clinical and sociodemographic variables (N = 193).

Variable	Adequate HL		Inadequate or Limited HL		
	*n* = 112		*n* = 81		
	*N*	%	*n*	%	*p*-Value
CATCHMENT AREA					
Ribera	83	57.2	62	42.8	0.828
Xativa-Ontinyent	29	60.4	19	39.6	
MARITAL STATUS					
Single-divorced-separated	36	48.6	38	51.4	0.037 *
Married	76	63.9	43	36.1	
LEVEL OF EDUCATION					
No studies or Primary Ed.	20	34.5	38	65.5	<0.001 *
VET	13	41.9	18	58.1	
HNC	32	65.3	17	34.7	
Bachelor’s degree	24	82.8	5	17.2	
University degree	23	88.5	3	11.5	
OCCUPATION					
Salaried	74	64.9	40	35.1	0.041 *
Unemployed	21	43.8	27	56.2	
Others (Student, SE)	17	54.8	14	45.2	
SECTOR					
Administration	30	69.8	13	30.2	0.008 *
Unemployed	26	41.9	36	58.1	
Healthcare	17	77.3	5	22.7	
Services	11	52.4	10	47.6	
Others (Industry, Education)	26	66.7	13	33.3	
COUNTRY OF ORIGIN					
Spain	106	59.9	71	40.1	0.082
Other	6	37.5	10	62.5	
PARTNER’S OCCUPATION					
Entrepreneur	15	71.4	6	28.6	0.305
Salaried worker	90	57.3	67	42.7	
Others	7	46.7	8	53.3	
DELIVERY TYPE					
Spontaneous	56	55.4	45	44.6	0.201
Instrumental	30	69.8	13	30.2	
C-section	26	53.1	23	46.9	
SMOKER					
Yes	24	42.1	33	57.9	0.004 *
No	88	64.7	48	35.3	
PREGNANCY RISK					
Low risk	96	62.3	58	37.7	0.016 *
High risk	16	41.0	23	59.0	

* *p* < 0.05 Chi-Square or Fisher’s exact tests. HL: Health Literacy by Newest Vital Sign; VET: Vocational Education and Training; HNC: Certificate of Higher Education (HNC); SE: Self-employed.

**Table 5 ijerph-17-02910-t005:** Multivariate logistic regression model for tobacco use during pregnancy (*N* = 193).

Variable	Variable	OR	95% CI	*p*-Value
HEALTH LITERACY	Adequate	1		
	Limited or inadequate	2.39	(1.24–4.63)	0.010 *
AGE FIRST PREGNANCY		0.94	(0.88–0.99)	0.027 *
CATCHMENT AREA	Ribera	1		
	Xativa-Ontinyent	0.28	(0.11–0.71)	0.008 *

* *p* < 0.05.

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
