# Peer review of "Influence of Puerperal Health Literacy on Tobacco Use during Pregnancy among Spanish Women: A Transversal Study"

_ijerph, 2020, doi:10.3390/ijerph17082910_

Round 1

Reviewer 1 Report

This is a paper on smoking and health literacy during pregnancy, but I have some comments to the author.

Comment 1. The description in the table is very difficult to read. It should be more readable, such as left-justifying Variables.

Comment 2. There are some places where the numbers (p-values) in the table and the text do not match. For example, Table 2 and Line 162-169.

Comment 3. I think that the description “We observed that tobacco consumption was related to low HL (P = 0.004)” in Line 166-167, is not appropriate expression. The author used Chi-square test in this analysis, and correlation cannot be elucidated by Chi-square test. Author can say “the prevalence of smoker was significantly higher in low HL group (P = 0.004)” or “low HL was significantly frequent in smoker group (P = 0.004)”, of course.

Author Response

Thank you.

Reviewer 2 Report

The study aims to analyse the relationship between the degree of Health literacy (HL) of women during the early puerperium and tobacco use during pregnancy. A multicentre, descriptive, cross-sectional study was carried out. Their HL level was obtained using the Newest Vital Sign (NVS) tool. The author found that a low level of HL is strongly associated with tobacco use, adjusted by marital status, education level, health care area and maternal age. Therefore they concluded that a low level of HL during the puerperium is related to tobacco consumption during pregnancy. The findings provide a preliminary support for HL as an independent predictor of tobacco consumption during pregnancy and could be an important factor in understanding health disparities related to tobacco use. HL should be considered in interventions related to the prevention of smoking during pregnancy. However, this study is merely an analysis of the association between HL and tobacco use without detailed and in-depth discussion. A major revision is needed for the manuscript before its publication.

My comments are as follows:

  1. The participants were sampled with a method of purposive sampling within two hospitals, which may confound this result of the study. The districts (urban, rural…), the reasons and process of purposive sampling should be detailed further in the Section of Methods. Otherwise this should be put into the limitation of the study.
  2. In the model, there is a lack of considering an important characteristic-- economic status (income), which may affect both HL and tobacco use, since health concept and behavior are closely related to education, economic status and occupation.
  3. There are conflicting results in Table 3 and 4. In Table 3, higher education level seemed reversely related to tobacco use (P<0.05). However this is not the case in Table 4 (OR 2.44 for tobacco use). Please clarify.
  4. As mentioned above, in the section of discussion the authors should add detailed and in-depth discussion for their observations, as well as the possible interventions. The authors may consider adding some descriptions for discusson since they have reported the results.
  5. The style of references is incorrect. Please make changes for your references to conform to the journal’s style.

Author Response

Thank you.

Reviewer 3 Report

This paper shows multiple substantial methodological and statistical flaws, most of which should be correctable by re-analysis and re-presentation of the currently available data set.

This study, in both title and text seemed to use the terms "tobacco use" and smoking as if they were synonymous. This should be clarified in the introduction or methods section, noting that tobacco smoking was the only behavior being considered.  Whether or not data were gathered as to marijuana use or use of other tobacco-related products (e-cigarettes, snus, chewing tobacco, cigars, water pipes, etc) -- should be noted. If interviewers and authors assumed that all such tobacco use was smoking, that too, should be noted.

One trivial flaw appears in the introduction section, with references 9 and 10.  Nicotine is not detectable in Urine.  Authors should re-check the references, as they should show either "cotenine" or "nicotine metabolites."

Lines 90-94 describe women excluded from the study on the basis of disabilities that are presumed to make them inadequate as to health literacy. Even though they were not subjected to the interview, providing their respective profiles, pregnancy outcome and smoking-related data would be a significant improvement to the paper, if such data are available.

Some of the most important data are those presented in lines 135-148.These data should be presented in tabular form in a separate table, in tabular form, by level of HL.

Table 2 has a number of significant problems.  First, the labels should be "Adequate HL," "Limited HL" and "Inadequate HL," with footnoted noting what each of these labels mean.  If there are no substantial differences between the Limited an Inadequate categories in terms of the critical issues addressed in this paper, this should be noted in the narrative. 

The most substantial error in Table 2 is the way in which the percentages were calculated.  There should be a row showing the total number of persons in the "Adequate" and "limited/Inadequate" categories, with no percentages. Then, for all the other parameters, the percentage should be the percentage based on the number of persons in that HL category.

This same error in calculating percentages is present in Table 3, which should be re-labeled as profile by smoking status.

A separate table should be shown for smokers, with the percentage of smokers in each HL category presented as a percentage of quitters within each HL category.

There should be some discussion as to the relationship of HL with pregnancy risk and outcomes and whether smokers who did not quit showed poorer outcomes.  Such a finding would significantly increase the interest in healthcare providers screening pregnant women for HL.

The discussion should highlight the finding that health literacy does not necessarily reflect what would otherwise be considered general literacy, given the numbers of women with limited educational attainment showing adequate health literacy and the finding that some women with high educational attainment showing limited or inadequate HL.

Finally, there should be a section at the end of the Discussion describing the limitations of this study.  The major limitations are that the data were gathered after the conclusion of the pregnancy and that they were not verified by laboratory study such as urine cotenine or by carbon monoxide testing. It should also be noted that bias may have been introduced by the general knowledge by participants that smoking during pregnancy was something that was frowned upon by the interviewers.

Author Response

Thank you.

Reviewer 4 Report

Influence of Puerperal Health Literacy on Tobacco 2 Use during Pregnancy among Spanish Women: A 3 Transversal Study.

The objective of this study was to examine the relationship between the degree of health literacy (HL) of women during the early puerperium and tobacco use during pregnancy.

In general the concept of the manuscript is of a great interest and the readership of the Journal may benefit from the presented facts and study results. The paper is well written and clear. The topic of the manuscript is interesting and not well studied, especially the usage of Newest Vital Sign (NVS) tool, which assess the health literacy.

I have some comments:

  1. The first problem is with the tool using to assess the health literacy. No information is included if this tool is really good or valid in assessment health literacy among pregnant women in certain age. This should be added.
  2. How the two hospitals from which the women were recruited were selected? Based on what kind of criteria?
  3. In my opinion the information (or the calculation) of the sample size is not really needed in the paper. It should be delated.
  4. Why the information about smoking was based only on electronic health records and the information received during pregnancy? Why during performing health literacy assessment such questions were not included? In my opinion after delivery the women are more willing to cooperate and this may reduce the underestimation of smoking which occurred during pregnancy.
  5. In the limitations it should be mention that no assessment of smoking (measuring the cotinine in urine) was performed.
  6. In the Conclusion section it should be mentioned how HL can help to reduce smoking among pregnant women.

Author Response

Thank you.

Round 2

Reviewer 2 Report

The authors' response and revision have addressed most of my concerns.

I think the manuscript can be sent for publication after listing "a lack of economic status" as a major limitation of the study.

Author Response

Thank you for the comment. As we stated in the limitations section, the economic status was not available as a variable, but we measured the education level, occupation and activity, which are also socioeconomic status indicators in line with other authors.

Thanks for your understanding.

Reviewer 3 Report

The revised manuscript is a major improvement over the original, but the statistics and presentation of the statistical data are still a mess.  With the changes that have been made, however, I have a much better understanding of what, in fact, you have found.

As I understand this paper, you are trying to show that screening for Health Literacy is something clinicians should do when dealing with pregnant women, since such screening will enable clinicians to do a better job helping smokers quit. Since lots of other variables are predictors of smoking status, these have to be considered when determining if HL is a variable not easily predicted by things like low educational attainment and low income. 

Furthermore, there are two smoking related variables to be considered. One is whether or not the woman smoked when entering pregnancy. The second is whether or not she quit during pregnancy. With that in mind, all of the tabular and narrative data in the paper should be formatted with that in mind; ie. clearly showing which are the confounding variables (marital status, education level, etc) as opposed to the dependent variables of major interest in this study. (smoking status and quitting).

In terms of the tabular data, I suggest that you merge Tables 1 and 5 into a new Table 1, with the rows by variable, and the columns by Smoker yes, no, percentages of each and row totals, with the percentages calculated by the row totals. To the far right you can show the row-specific p values. HL as a variable should be shown at the very end.

Table 2 is fine as it is.

Table 4, dealing with quitters, then becomes Table 3.  Rows and columns should be reversed so the columns are quits, not quits, and total, and the rows Adequate HL and inadequate or Limited HL, with the percentages calculated by row.   Herein lies a major rub.  The narrative says that quitting did not vary by HL, but the table seems to show that it does.  The correct answer is that not only is the difference not statistically significant, to the degree that there is a difference, a greater percentage of low HL women quit than high HL women.  Since the clinician has no control over initial smoking status, and can only influence quit rates, this finding seems to indicate that screening for HL is of no practical value in identifying women in need of extra counseling to help them quit.

With regard to the subsequent tables, you need to take the variables showing significance in smoking status (catchment area, married, unemployed and high risk pregnancy) and show the data that would tell whether HL is an independent variable or simply reflects one or more of the others.  Since the difference by catchment area is so great, you need to see if that variable simply reflects education and employment.

In the abstract you note adjustment for a number of variables, including maternal age, but your data show no difference by maternal age.  This has to be either better addressed in the results and discussion, or deleted from the abstract.

The major issue you  need to deal with will by why HL is worth screening for if it will not help identify those needing more health education for smoking cessation.

Author Response

Thank you.

Reviewer 4 Report

I have no further comments

Author Response

Thank you for your assistance.

Round 3

Reviewer 3 Report

The authors are to be commended for their responsiveness and speed of response.  

Quality of presentation and scientific soundness should be changed from "low" to "high."  

The overall recommendation is now "accept after minor revision"   The revisions suggested are as follows:

1.  There appears to be a typographical error at the end of line 141.  It should read <0.05.  

2.  Table 3 and the smoking cessation data require a bit more attention. I was unable to figure out the denominator(s) for the percentages shown in the table.  As I read it, 40 quit and 17 did not (the figures shown are mis-labled). Of the 24 women with Adequate HL, 16 (67%) quit. Of those with inadequate or limited HL a larger percentage quit (34 of 33 or  72%). While this difference was statistically significant, it still left a higher percentage of the Inadequate/Limited group smoking at the end of the pregnancy.   

3. There should be a bit more narrative in the paragraph lines 209-214  noting that despite this higher quit rate,more lower HL women smoked thru the entire pregnancy. You may want to note the number and percentage of women in each HL group who were still smoking at the end of the pregnancy. This would probably be best in a narrative discussing the data in Table 3.  

4. I have no idea what the sentence re goal, lines 221-222 means. This should be explained in more detail or deleted.  

5. Finally, it seems to me that with these clarifications, the authors could strengthen the conclusion and strengthen the interest in this paper by noting that, while HL was associated with levels of education and markers of social class, there were enough women whose HL score was not what was predicted by these variables to make screening for HL in pregnancy worthwhile.

Author Response

Thank you.
